# LLaVA-PruMerge: Adaptive Token Reduction for Efficient Large Multimodal Models

## Abstract

Large Multimodal Models (LMMs) have shown significant visual reasoning capabilities by connecting a visual encoder and a large language model. LMMs typically take in a fixed and large amount of visual tokens, such as the penultimate layer features in the CLIP visual encoder, as the prefix content. Recent LMMs incorporate more complex visual inputs, such as high-resolution images and videos, which further increases the number of visual tokens significantly. However, due to the inherent design of the Transformer architecture, the computational costs of these models tend to increase quadratically with the number of input tokens. To tackle this problem, we explore a token reduction mechanism that identifies significant spatial redundancy among visual tokens. In response, we propose `PruMerge`, a novel adaptive visual token reduction strategy that significantly reduces the number of visual tokens without compromising the performance of LMMs. Specifically, to metric the importance of each token, we exploit the sparsity observed in the visual encoder, characterized by the sparse distribution of attention scores between the class token and visual tokens. This sparsity enables us to dynamically select the most crucial visual tokens to retain. Subsequently, we cluster the selected (unpruned) tokens based on their key similarity and merge them with the unpruned tokens, effectively supplementing and enhancing their informational content. Empirically, when applied to LLaVA-1.5 [Liu et al., 2023a] and Video-LLaVA [Lin et al., 2024], our approach can reduce the number of visual tokens by 4 times, and achieve comparable or better performance across diverse visual question-answering and reasoning tasks.

## 1 Introduction

Large Language Models (LLMs) [OpenAI, 2023b, Team et al., 2023, Jiang et al., 2023, Touvron et al., 2023] have shown strong reasoning abilities. LLMs are usually high-capacity Transformers [Vaswani et al., 2017] pretrained with a large-scale text corpus. Large Multimodal Models (LMMs), inherit LLMs for text generation, while also leveraging a visual encoder such as CLIP-ViT [Radford et al., 2021] to embed image patches into visual tokens as the prefix visual context.

LMMs need substantial computation for inference. The LLM is the primary factor for the high computation cost, since the visual encoder is usually quite small relative to the LLM. For example, the commonly used CLIP visual encoder, ViT-L, only has 0.3B parameters, while the corresponding LLM such as LLaMA [Touvron et al., 2023] or Vicuna [Vicuna, 2023] can have 7B or 13B parameters. As a result, reducing the LLM's inference cost is the key to achieving low LMM inference cost.

Prior works [Chu et al., 2023; 2024, Yuan et al., 2023a] mainly focus on replacing the LLM backbone with a smaller language model with less parameters, such as Phi-2 [Javaheripi et al., 2023]. However, such approaches sacrifice the reasoning abilities of LLMs, leading to a large performance gap on visual question-answering and reasoning tasks such as VQAv2 and MM-Bench [Chu et al., 2024]. A similar approach is to apply quantization for LLMs [Liu et al., 2023b, Yuan et al., 2024].

However, the cost of LLMs comes from not only its large number of parameters, but also the *length of the input context* due to the quadratic complexity of the Transformer's attention operation. The context length in LMMs is especially important, where a fixed amount of visual tokens serves as the prefixed tokens. For example, in LLaVA-1.5, 576 visual tokens are appended, and in Video-LLaVA [Lin et al., 2024] that number is even higher, leading to high training and inference costs.

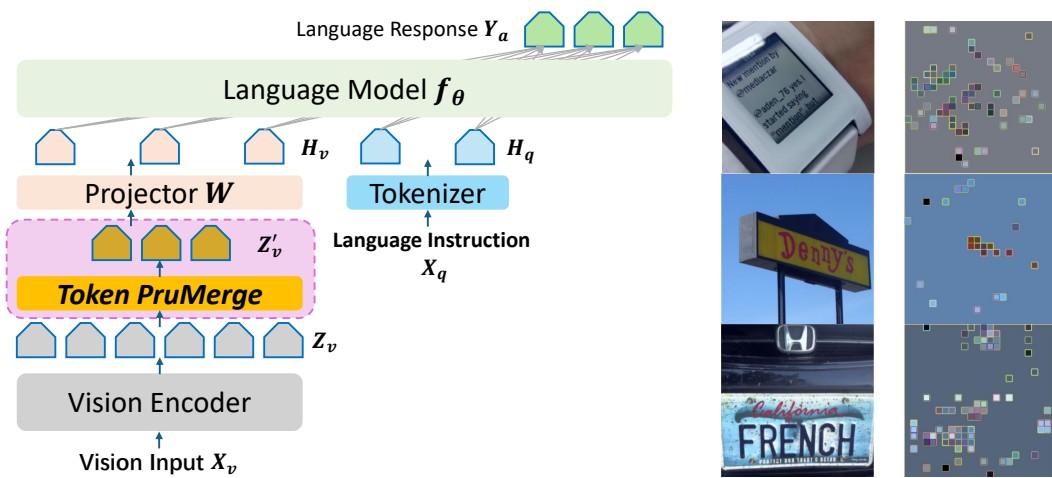

(a) Main idea of **PurMerge**.  (b) PruMerged Token Visualization.

Figure 1: **(a)** We prune and merge visual tokens produced by the vision encoder, while keeping all other procedures of the LMM the same. By reducing the number of visual tokens, `PruMerge`, significantly reduces the computation cost for text generation in LMMs (around 4-10 times in FLOPs for LMM prefill), while can maintain comparable performance. **(b)** A visualization of the selected tokens. `PruMerge` can **adaptively** select visual tokens based on the information density of the visual input, enabling the LLM to perceive visual input effectively and efficiently. More attentive tokens are sampled in complex images such as ones with text, while fewer are sampled on simpler images. The attentive tokens are usually located at the regions with dense information.

Thus, an intriguing question is: *Can we reduce the number of prefix visual tokens while maintaining comparable performance?*

In our study, we find that many visual tokens are redundant, similar to findings in previous related work [Bolya et al., 2023, Liu et al., 2022], and most of the visual tokens can be pruned with little sacrifice in performance. In particular, the similarity (i.e., attention scores in the visual encoder's self-attention module) between the class token and spatial patches are sparse, indicating that only a small number of visual tokens are related to key visual information in the visual samples. Motivated by this, we use this sparse similarity to adaptively select important visual tokens, as shown in Fig.1b. Specifically, we leverage the Interquartile Range (IQR) [Boukerche et al., 2020] scoring function in outlier detection to prune unimportant visual tokens. Moreover, we merge the visual tokens using $k$-nearest neighbor and update the selected important visual tokens via weighted averaging, which further enhances performance. Finally, we design `PruMerge+`, which samples visual tokens spatial-uniformly to complement the unpruned tokens. `PruMerge+` not only minimizes performance degradation but also ensures substantial token reduction, maintaining a more comprehensive and representative selection of visual tokens.

Empirically, `PruMerge` can effectively and adaptively reduce the visual tokens in each image in LLaVA-1.5 [Liu et al., 2023a], where with just 5.5% of visual tokens, which is around 32 tokens for an image on average, LLaVA-PruMerge can maintain comparable performance with that of retaining all 576 tokens across diverse benchmarks. Furthermore, `PruMerge` showcases its versatility across various modalities, including video. By integrating `PruMerge` with Video-LLaVA [Lin et al., 2024] during the inference phase alone (*i.e.*, no need for additional training) we not only expedite processing within video-LLMs but also enhance their performance across multiple benchmarks.

## 2  RELATED WORK

### 2.1  EFFICIENT LARGE MULTIMODAL MODELS (LMMS)

Large Language Models (LLMs) such as GPT-4 [OpenAI, 2023b], LLaMA [Touvron et al., 2023], Mistral [Jiang et al., 2023], and Gemini [Team et al., 2023] have demonstrated strong question answering and reasoning capabilities over text. Large Multimodal Models (LMMs) [Liu et al., 2023b, Zhu et al., 2023, Yin et al., 2023, Zhang et al., 2024] extend these reasoning capabilities to images,

where given an image and an associated question, a vision encoder and an LLM are leveraged to generate text responses in a chat format. More recent works extend whole-image understanding into region-level understanding [Cai et al., 2024, Zhang et al., 2023b, Peng et al., 2023, Chen et al., 2023], video understanding [Lin et al., 2024, Zhang et al., 2023a] and 3D scene understanding [Hong et al., 2023]. Such works typically feed the visual tokens directly into the LLM as prefix tokens, via either an MLP [Liu et al., 2023a], Qformer [Dai et al., 2023, Zhu et al., 2023], or resampler [Alayrac et al., 2022]. The number of visual tokens can be prohibitively long, especially when the images are high-resolution [Liu et al., 2024, OpenAI, 2023a]. In this paper, we reduce the number of visual tokens with a novel adaptive prune and merge procedure.

While LMMs have made significant advances, their large-scale training and deployment incur significant computational costs, requiring efficient parallel device implementations. Google's Gemini [Team et al., 2023] is a pioneer in efficient LMMs, achieving state-of-the-art performance on multimodal benchmarks and introducing mobile-scale LMMs suitable for low-memory devices, although it is not open-source. Open-source alternatives like LLaVA-1.5 [Liu et al., 2023a] employ advanced compression techniques such as 4/8 bit quantization [Dettmers et al., 2022, Shang et al., 2024]. MobileVLM [Chu et al., 2023] and its improved version, MobileVLM-v2 [Chu et al., 2024], focus on compact architecture designs and training optimizations for mobile use.

In most cases, LMM efficiency is enhanced by reducing the size of the backbone of the LMM, but no work has considered the efficiency of the LMM from the perspective of the number of visual tokens.

## 2.2 TOKEN REDUCTION

The quadratic complexity of Transformers [Vaswani et al., 2017] poses a significant challenge in scaling input sequence length. Various approaches try to address this issue. Sparse attention methods, e.g., Linformer [Wang et al., 2020] and ReFormer [Kitaev et al., 2020], reduce complexity by limiting attention operations to specific regions rather than the full context. Token reduction can also accelerate Transformers [Haurum et al., 2023]. Methods like [Liu et al., 2022, Yin et al., 2022, Liang et al., 2022, Bolya et al., 2023, Fayyaz et al., 2022] focus on reducing the number of tokens within the internal transformer structure, thereby decreasing computational load. For instance, token merging [Bolya et al., 2023] employs full attention but progressively reduces tokens in each transformer block by selecting the most representative tokens through bipartite matching. However, these **uni-modal token reduction methods** are not directly applicable to LMMs. One of the main inefficiencies in LMMs stems from their use of numerous prefix visual tokens as a fixed context budget [Liu et al., 2023b, Zhu et al., 2023] (analyzed further in Sec. 4.2), not from the internal structure of Transformers. We discuss the unsuitability of existing uni-modal token reduction methods for LMM acceleration in Sec. 3.5. In our study, we introduce a plug-and-play token reduction method specifically designed for LMMs. Our approach, based on visual token similarities, achieves comparable performance while using less than one-fourth of the original tokens. The core of our method is a sparsity-based selection mechanism that identifies "anchor" tokens via sparse attention scores within the modality encoder, and is the most crucial design element of `PruMerge`. In parallel to our work, Shi et al. [2024] proposes CrossGet, a graph-matching-based algorithm for token matching. While both approaches aim to reduce tokens in multimodal contexts, they differ significantly in their methodologies. Beyond the token selection module, our token merging module also differs from CrossGet's graph soft matching. Our k-nearest neighbors clustering approach has a time complexity of $O(n)$, which is more computationally efficient compared to CrossGet's $O(n^2)$ complexity, thus enhancing scalability and efficiency.

## 3 METHOD: TOKEN PRU-MERGING

In this section, we first review the basic implementation of large mutilmodal models (LMMs), with a particular focus on the visual encoder component (*i.e.*, Vision Transformer). We highlight the direct correlation between the number of visual tokens and the efficiency of LMMs (Sec. 3.1). Next, we present a plug-and-play token reduction method specifically designed for LMMs, called token `PruMerge`. Our method features two key components: (1) Adaptive Important Token Selection (AITS) via Outlier Detection which adaptively determines the optimal number of visual tokens to retain based on the unique characteristics of the image (Sec. 3.2); and (2) Token Supplement (TS) via Similar Key Clustering, which facilitates efficient processing without compromising the model's performance by maintaining the integrity and richness of the visual information (Sec. 3.3).

## 3.1 PRELIMINARIES

**Vision Transformers (ViTs)** [Dosovitskiy et al., 2020] are the most widely used vision encoder for LMMs, in which the input image is converted into a sequence of representative tokens by the ViT, and then fed into an LLM for understanding [Liu et al., 2024, Zhu et al., 2023, Hong et al., 2023, Zhang et al., 2024]. An input image is divided into a grid of patches and each patch is projected into a token embedding by the ViT. In addition to the patch tokens, a class token (*i.e.*, `[CLS]` token) is computed to aggregate global image information for classification. A ViT consists of a set of transformer blocks, which in turn consist of several essential components: a multi-head self-attention (MSA) layer, a feed-forward neural network (FFN), skip connections, and layer normalization [Ba et al., 2016]. These components work together to improve the model's capability to understand visual data [Han et al., 2022]. In the self-attention layer, an input token is projected into three distinct vectors: query $\mathbf{q}$, key $\mathbf{k}$, and value $\mathbf{v}$, using three linear transformation matrices $\mathbf{W}_q$, $\mathbf{W}_k$, and $\mathbf{W}_v$. These vectors, corresponding to different inputs, are assembled into matrices $\mathbf{Q}$, $\mathbf{K}$, and $\mathbf{V}$, respectively. The self-attention computes the relevance of each item to other items:

$$\mathbf{Y} = \text{Self-Attention}(\mathbf{Q}, \mathbf{K}, \mathbf{V}) = \mathbf{A} \cdot \mathbf{V} \tag{3.1}$$

where attention matrix $\mathbf{A} = \text{softmax}\left(\frac{\mathbf{Q} \cdot \mathbf{K}^{\mathbf{T}}}{\sqrt{d_k}}\right)$ and $d_k$ is the dimension of $\mathbf{q}$ and $\mathbf{k}$. In the last layer of the ViT, the `[CLS]` token is used for classification. Similarly, the attention between `[CLS]` token and other visual tokens is computed by the attention mechanism:

$$\mathbf{a}_{\texttt{cls}} = \text{softmax}\left(\frac{\mathbf{q}_{\texttt{cls}} \cdot \mathbf{K}^{\mathbf{T}}}{\sqrt{d_k}}\right). \tag{3.2}$$

The MSA framework allows for simultaneous attention on multiple positions, offering diverse representation subspaces. This is achieved by employing distinct query, key, and value matrices for different heads, which project the input vectors into different representation subspaces. After the self-attention layers is the feed-forward network (FFN), which consists of two linear transformation layers separated by a nonlinear activation function:

$$\text{FFN}(\mathbf{X}) = \mathbf{W}_2 \sigma(\mathbf{W}_1 \mathbf{X}) \tag{3.3}$$

where $\mathbf{W}_1$ and $\mathbf{W}_2$ are the matrices of the linear transformation layers, and $\sigma$ denotes the nonlinear activation function. The general forward pass of ViT is illustrated in the left part of Figure 2.

**Large Multimodal Models (LMMs)**. Following the forward pass through a Vision Transformer (ViT), a set of visual tokens is generated. These tokens are then processed by the input projector $\mathbf{\Theta}_{\mathbf{X} \rightarrow \mathbf{T}}$, which maps the encoded visual features from $\mathbf{F}_X$ into the text feature space $\mathbf{T}$. The aligned features and the text prompts $\mathbf{P}_T$ are then fed into the LLM backbone [Zhang et al., 2024]. The overall architecture of an LMM is depicted in Figure 1.

Importantly, the computational cost with these models increases quadratically with the number of input tokens to the LLM [Tay et al., 2022]. Mathematically, if there are $N$ tokens in the input, the self-attention mechanism computes a $N \times N$ matrix of attention scores, where each entry in this matrix represents the attention score between a pair of tokens. However, there is an increasing demand for processing high-resolution images and videos, which increases the number of visual tokens, further exacerbating computation costs. The reduction of visual tokens presents a promising approach to improving the efficiency of LMMs by reducing the escalating computational requirements.

## 3.2 ADAPTIVE IMPORTANT TOKEN SELECTION VIA OUTLIER DETECTION

The most straightforward solution to improve the efficiency of visual token utilization in LMMs is to prune redundant visual tokens [Liu et al., 2022, Yin et al., 2022, Liang et al., 2022]. To realize token pruning, we need to address a pivotal question: *How do we determine the importance of each visual token?*

As discussed in Sec. 3.1, LMMs typically leverage an extensive stack of visual tokens to represent the visual information. On the other hand, self-/weakly-supervised learning paradigms, such as CLIP [Radford et al., 2021] simplify this complexity by representing an entire image with a single `[cls]` token, regarded as the most information-condensed token. To balance those two extreme

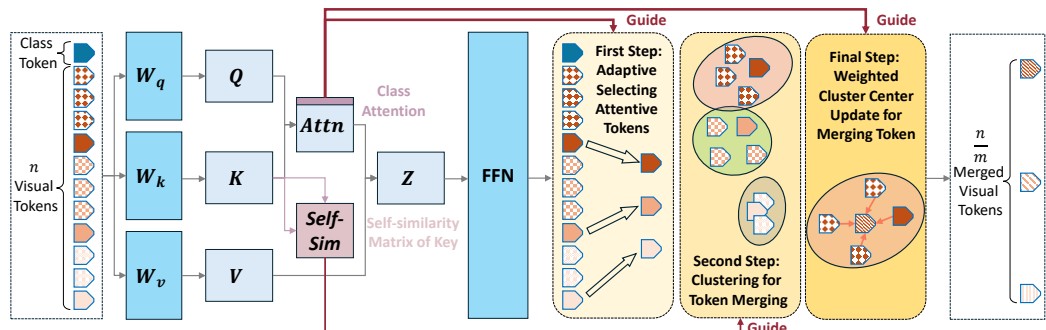

Figure 2: `PruMerge` has 3 steps: (1) Sample important tokens according to the similarities between the class tokens and spatial visual tokens via an outlier detection algorithm (see Sec.3.2); (2) Cluster the visual tokens via $k$-nearest neighbor; and (3) Adjust the sampled visual tokens via weighted averaging for each cluster (see Sec.3.3). Here $m$ denotes the visual token compression ratio.

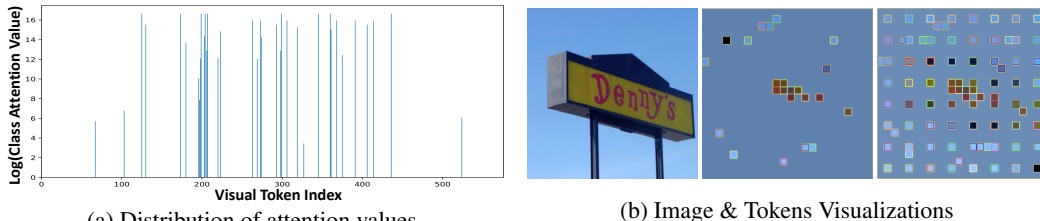

(a) Distribution of attention values.    (b) Image & Tokens Visualizations

Figure 3: (a) Distribution of attention scores (in CLIP-ViT's penultimate layer) between the [cls] token and visual tokens. The y-axis shows logarithmic values. Notably, most spatial visual tokens have near-zero attention values with the class token. (b) Visualizations of `PruMerge` and `PruMerge+`.

paradigms, we investigate the Key-Query attention between [cls] token and visual tokens, *i.e.*, $\mathbf{a}_{cls}$ in Equation 3.2. Observing the distribution patterns of attention between the [cls] token and visual tokens unveils a sparse landscape, as depicted in Figure 3a. This sparse distribution underpins our methodology for identifying crucial visual tokens. By employing outlier detection algorithms, we aim to adaptively select visual tokens that best represent an image's features effectively.

**Interquartile Range (IQR) Method for outlier detection**. To identify outliers within class attention values, we adopt the Interquartile Range (IQR) method [Boukerche et al., 2020], a statistical technique known for its robustness in outlier detection. Its essence lies in its capability to establish a boundary or "fence" that delineates the normal range of data. This is achieved by calculating the IQR (the difference between the third quartile Q3 and the first quartile Q1) and subsequently defining the outer limits of the normal range as 1.5 times the IQR above Q3 and below Q1. Specifically, the computation is as follows: the "lower fence" is set at $1.5 \times$ IQR below Q1, and the "upper fence" is set at 1.5 $\times$ IQR above Q3. Any attention values residing outside these fences are classified as outliers. In practice, only the "upper fence" is activated since the attention score is positive. Through this method, we can adaptively identify and select the visual tokens for each image that exhibit outlier attention values, *i.e.*, those playing a significant role in representing the image within the LMM context. Note that we use the class attention value from the penultimate layer for this calculation.

As shown in Figure 1b, the sampled visual tokens demonstrate two behaviors: (1) The number of attentive tokens are proportional to the complexity of the image. Simpler images such as "*Billboard among blue sky*" owns fewer tokens while images with rich information such as a screen with dense texts own more tokens. (2) The sampled tokens are typically spatially aligned with important content. Such visualizations align with our visual token sampling design. These trends are also observed at the benchmark level; in Table 4, the average token numbers on various benchmarks differ.

### 3.3 TOKEN SUPPLEMENT VIA SIMILAR KEY CLUSTERING

Following the selection of informative visual tokens, we next optimize the utilization of the remaining tokens. While pruned tokens may initially seem extraneous, they hold potential value for the

**Algorithm 1** Token `PruMerge` and `PruMerge`+ algorithms for reducing the number of visual tokens.

---

**Require:** Key and Query matrices of ViT's penultimate layer, $\mathbf{K} = \{\mathbf{k}_1, \cdots \mathbf{k}_n\}$ and $\mathbf{Q} = \{\mathbf{q}_1, \cdots \mathbf{q}_n\}$. The penultimate layer's output tokens, $\mathbf{Y} = \{\mathbf{y}_1, \cdots \mathbf{y}_n\}$. $n$ is the number of input visual tokens.

**Ensure:** Refine $\mathbf{Y}$ to $m$ (adaptive) visual tokens $\mathbf{Y}' = \{\mathbf{y}'_1, \cdots \mathbf{y}'_m\}$, in which $m \ll n$.

1: **Token `PruMerge`:**
2: Calculate attention between visual token and class token $\mathbf{a}_{[cls]}$ using Equation 3.2.
3: Use the outlier detection algorithm IQR to **adaptively** select $m$ important visual tokens' indices $\{i_1, \cdots, i_m\}$ based on $\mathbf{a}_{[cls]}$ (see Sec. 3.2).
4: (Optional for `PruMerge`+, see Sec. 3.4) Calculate the outlier ratio $r_o = \frac{m}{n}$.
5: (Optional for `PruMerge`+) Spatial-uniformly sample visual tokens with $r_o$, and get $\{i_{1+m}, \cdots, i_{2m}\}$.
6: (Optional for `PruMerge`+) Update the selected tokens' index with $\{i_1, \cdots, i_m, i_{m+1}, \cdots, i_{2m}\}$.
7: **for** $p = \{i_1, \cdots, i_m\}$ **do** (see Sec. 3.3)
8:     Calculate the distance between selected token $\mathbf{y}_p$ and other visual tokens, $\mathbf{y}_{\{1, \cdots, n\}/p}$;
9:     Use $\mathbf{y}_p$ as cluster center and find the $k$ most similar tokens, with indices $\{j_1, \cdots, j_k\}_p$;
10:    Update cluster center token with weighted sum: $\mathbf{y}'_p = \sum_{q=1}^{k} \mathbf{a}[j_q] \cdot \mathbf{y}_{j_q}$;
11: **end for**
12: Output a refined stack of visual tokens $\mathbf{Y}' = \{\mathbf{y}'_1, \cdots \mathbf{y}'_m\}$.

---

perception capabilities of the LLM backbone. This potential arises particularly in cases where an image contains large object parts that dominate the scene. In such scenarios, overly aggressive pruning could inadvertently diminish the model's ability to represent the image comprehensively.

To address this, we devise a token merging method aimed at enhancing the representational capacity of the selected unpruned tokens. This method involves the strategic fusion of currently pruned tokens, as depicted in Figure 2. To choose the pruned tokens to merge, we need a way to measure similarity between visual tokens. Here we leverage the self-attention mechanism in ViTs. Since the key vector of each patch token already contains information summarized in the self-attention module [Vaswani et al., 2017], the final layer's key vector serves as the representation. And then we use the dot product between keys to calculate which tokens have similar visual information [Bolya et al., 2023]:

$$\text{Sim}(\mathbf{y}_i, \mathbf{y}_j) = \mathbf{k}_i \cdot \mathbf{k}_j^T, \tag{3.4}$$

which yields $\mathbf{KK}^T(i, j)$ for tokens $i, j$ in vectorized form for the set of all tokens $1, 2, \cdots, n$, where $n$ is the number of input visual tokens.

With the similarities between visual tokens established, we simply find the $k$-nearest neighbors for each unpruned token, which act as the cluster centers. The integration of pruned tokens into these clusters is guided by their respective class attentions $\mathbf{a}[i]$, enabling a refined representation of each unpruned token through a weighted sum. This procedure is outlined in Algorithm 1.

## 3.4 PRUMERGE+: BRIDGING THE EFFICIENCY-PERFORMANCE GAP

While `PruMerge` achieves a remarkable reduction in the number of visual tokens—over tenfold compared to the original setup—the process is not without drawbacks. Specifically, the compression technique, though efficient, introduces a marginal performance discrepancy between the original LLaVA model and its `PruMerge`-optimized counterpart, LLaVA-PruMerge. To address this, we introduce `PruMerge`+, a refined version that strikes an optimal balance between token reduction and model performance.

`PruMerge`+ enhances our original method by maintaining the ability to significantly reduce visual token count—by an average of fourfold—with minimal performance degradation. This improvement is detailed in Algorithm 1, building upon the token selection strategies outlined in Section 3.2. A new aspect of `PruMerge`+ lies in its enhanced token selection process. Beyond merely focusing on the previously identified important tokens, `PruMerge`+ extends its reach to encompass additional visual tokens from areas initially deemed less critical. This is achieved through a spatially uniform sampling of visual tokens, guided by a predetermined ratio informed by the distribution of outlier tokens. This

Table 1: Comparison with large multimodal models on six benchmarks. Our `PruMerge` and `PruMerge`+ can adaptively reduce visual tokens, which use only (respectively) 5.5% and 25.0% visual tokens on average (on 6 tasks) and achieves competitive performance to the original LLaVA-1.5.

| Method | LLM | Res. | PT | IT | VQA$^{v2}$ | SQA$^I$ | VQA$^T$ | POPE | MME | MMB |
|--------|-----|------|-----|-----|-----|-----|-----|------|-----|-----|
| BLIP-2 | Vicuna-13B | 224 | 129M | - | 41.0 | 61 | 42.5 | 85.3 | 1293.8 | - |
| InstructBLIP | Vicuna-7B | 224 | 129M | 1.2M | - | 60.5 | 50.1 | - | - | 36 |
| InstructBLIP | Vicuna-13B | 224 | 129M | 1.2M | - | 63.1 | 50.7 | 78.9 | 1212.8 | - |
| Shikra | Vicuna-13B | 224 | 600K | 5.5M | 77.4 | - | - | - | - | 58.8 |
| IDEFICS-9B | LLaMA-7B | 224 | 353M | 1M | 50.9 | - | 25.9 | - | - | 48.2 |
| IDEFICS-80B | LLaMA-65B | 224 | 353M | 1M | 60.0 | - | 30.9 | - | - | 54.5 |
| Qwen-VL | Qwen-7B | 448 | 1.4B | 50M | 78.8 | 67.1 | 63.8 | - | - | 38.2 |
| Qwen-VL-Chat | Qwen-7B | 448 | 1.4B | 50M | 78.2 | 68.2 | 61.5 | - | 1487.5 | 60.6 |
| LLaVA-1.5 | Vicuna-7B | 336 | 558K | 665K | 78.5 | 66.8 | 58.2 | 85.9 | 1510.7 | 64.3 |
| LLaVA-1.5 + `PruMerge` | Vicuna-7B | 336 | 558K | 665K | 72.0 | 68.5 | 56.0 | 76.3 | 1350.3 | 60.9 |
| LLaVA-1.5 + `PruMerge`+ | Vicuna-7B | 336 | 558K | 665K | 76.8 | 68.3 | 57.1 | 84.0 | 1462.4 | 64.9 |
| LLaVA-1.5 | Vicuna-13B | 336 | 558K | 665K | 80.0 | 71.6 | 61.3 | 85.9 | 1531.3 | 67.7 |
| LLaVA-1.5 + `PruMerge` | Vicuna-13B | 336 | 558K | 665K | 72.8 | 71.0 | 58.4 | 78.5 | 1428.2 | 62.3 |
| LLaVA-1.5 + `PruMerge`+ | Vicuna-13B | 336 | 558K | 665K | 77.8 | 71.0 | 58.6 | 84.4 | 1485.5 | 65.7 |

methodology ensures a more comprehensive and representative selection of visual tokens, as depicted in Figure 3b, thereby minimizing performance losses while still achieving substantial token reduction.

## 3.5 DISCUSSION: DISTINCTION FROM EXISTING UNI-MODAL TOKEN REDUCTION METHODS

**Uni-Modal token reduction methods** [Liu et al., 2022, Yin et al., 2022, Liang et al., 2022, Bolya et al., 2023, Fayyaz et al., 2022, Haurum et al., 2023] primarily focus on accelerating ViT computation speed by progressively reducing token numbers across transformer blocks, thereby lowering the internal ViT computational cost. Our approach, however, differs in its primary objective. Rather than targeting ViT efficiency, we aim to enhance the overall efficiency of LMMs, where the ViT is just one component with relatively light computational cost, as shown in Fig. 1. This design offers two key advantages: First, while ViTs typically rely on a single class token to represent the input image, which enables them to maintain performance despite a reduction in intermediate tokens, LMMs usually require a large stack of visual tokens. This ensures a comprehensive representation of the visual content, preserving the model's ability to capture nuanced details. Thus, using previous token reduction methods to obtain one refined class token to represent the visual input is not consistent with the literature of LMMs. Second, considering that the bulk of computational demand within LMMs is attributed to the LLM component rather than the ViT, our approach focuses not only on the reduction of tokens but also on maximizing the informational content of the pruned visual tokens. This strategy addresses the computational challenges inherent in LMMs with minimal compromise in the quality of the visual representation. Indeed, experiments comparing `PruMerge` with implementations of uni-modal token reduction methods in LMMs demonstrate significantly better performance for our method (see Sec. 4.4.4).

## 4 EXPERIMENTS

We first show the empirical performance of our approach when applied to LLaVA-1.5 in Sec 4.1. We then analyze the efficiency improvement by using our `PruMerge` on LMM in Sec 4.2. To show the generalization ablity, we conduct a series of experiments in Sec. 4.3. Finally, we demonstrate the effectiveness of each component in our model in Sec 4.4.

### 4.1 MAIN RESULTS

We apply our method to LLaVA-1.5 [Liu et al., 2023a], a recent state-of-the-art LMM. We further finetune LLaVA-1.5 using LoRA [Hu et al., 2022] for 1 epoch using the LLaVA-1.5 instruction fine-tuning data [Liu et al., 2023a] with our reduced visual tokens.

We evaluate on diverse visual question-answering and reasoning benchmarks including VQAv2 [Goyal et al., 2017], ScienceQA [Lu et al., 2022], TextVQA [Singh et al., 2019], POPE hallucination bench [Li et al., 2023b], MME [Fu et al., 2023], and MMBench [Liu et al., 2023c]. As shown in Table 1, our approach achieves comparable performance with LLaVA-1.5 despite using only a small fraction of the visual tokens, and performing better than previous works such as BLIP2 [Li et al.,

Table 2: Computation Cost Analysis. The development device is Tesla V100 GPU, and time estimated by the roofline model represents the theoretical performance that the hardware can achieve.

| Method | LLM Backbone | Quantization | FLOPs (TB) | Prefill Time (ms) | Total Memory (GB) | Storing Activation (GB) |
|---|---|---|---|---|---|---|
| LLaVA-1.5 | Vicuna-7B | FP16 | 9.3 | 88.6 | 23.3 | 4.60 |
| LLaVA-1.5 w/ `PruMerge` | Vicuna-7B | FP16 | 0.91 | 15.3 | 13.7 | 0.28 |
| LLaVA-1.5 | Vicuna-7B | INT4 | 2.3 | 151.6 | 5.9 | 1.20 |
| LLaVA-1.5 w/ `PruMerge` | Vicuna-7B | INT4 | 0.28 | 14.9 | 3.5 | 0.07 |
| LLaVA-1.5 | Vicuna-13B | FP16 | 18.2 | 170.5 | 41.6 | 7.30 |
| LLaVA-1.5 w/ `PruMerge` | Vicuna-13B | FP16 | 1.80 | 29.5 | 26.6 | 0.44 |
| LLaVA-1.5 | Vicuna-13B | INT4 | 4.6 | 294.9 | 10.5 | 1.80 |
| LLaVA-1.5 w/ `PruMerge` | Vicuna-13B | INT4 | 0.45 | 29.0 | 6.8 | 0.11 |

2023a] and InstructBLIP [Dai et al., 2023]. Specifically, in POPE and ScienceQA, our approach even shows better performance than LLaVA-1.5. Note that due to the adaptive nature of `PruMerge` (see Sec. 3.2), the token numbers for various tasks are different (see 4.4.1), and thus we use the average number on numbers of 6 tasks (*i.e.*, 32) for simplicity.

## 4.2 EFFICIENCY ANALYSIS

To elucidate the computational efficiency afforded by `PruMerge`, we utilize the roofline-based LLM-Viewer analysis developed in [Yuan et al., 2024]. Our investigation is grounded in a theoretical scenario tailored to highlight the impact of `PruMerge` on processing efficiency within LMMs. Consider a typical scenario where an image of dimensions $336 \times 336$ pixels is processed using a CLIP-ViT model, resulting in 576 visual tokens. Accompanying this image is a text prompt, assumed to contain 40 tokens for the sake of this analysis. Through the application of `PruMerge`, we achieve a dramatic reduction in the number of visual tokens, decreasing the original count by approximately 14.4 times in MME/TextVQA to match the token count of the text prompt ($576/14.4 \approx 40$). The implications of this reduction are significant, as demonstrated in Table 2, which outlines the computational cost associated with the LMM prefill process. Notably, `PruMerge` not only enhances the speed of the LLM prefill process by reducing the required floating-point operations (FLOPs) but also contributes to a reduction in computational memory demands.

It is important to emphasize that the benefits of `PruMerge` extend beyond mere efficiency gains. Our token reduction strategy can complement other LLM acceleration techniques, such as quantization and factorization [Yuan et al., 2023b]. This orthogonal relationship underscores the versatile potential of `PruMerge` to contribute to a broader spectrum of efficiency-enhancing strategies.

## 4.3 GENERALIZATION ON VIDEO-LLM

To assess the generalization capabilities of `PruMerge` and `PruMerge+` across different modalities, we next extend our approach to Video-LLaVA [Lin et al., 2024]. Video-LLaVA is one of the most popular open-soruced Video-LLMs. We seamlessly integrate both algorithms into Video-LLaVA without the need for additional training, enabling us to bypass re-training on video datasets during inference. The outcome of this integration is shown in Table 3. Video-LLaVA samples 8 frames from a video clip and extracts $8 \times 16 \times 16 = 2048$ visual tokens using a visual encoder for LLM perception, which is 4 times of visual token than LLaAV-1.5 [Liu et al., 2023a]. Our Algorthms `PruMerge` and `PruMerge+` can adaptively select important 256 (12.5% on average) and 256 (25.0% on average) important visual tokens, respectively. The results demonstrate that our algorithms not only reduce the number of visual tokens in Video-LLaVA but also is able to enhance its performance. This finding is noteworthy as it suggests a significant redundancy in the visual tokens used by video-LLMs. Exploring ways to further capitalize on this redundancy could shape future research directions.

## 4.4 ABLATION STUDY

### 4.4.1 TOKEN SAMPLING STRATEGY ANALYSIS

Here we show how our approach performs better than the vanilla visual token sampling strategy, including sequential sampling and spatial sampling.

Table 3: Comparison of different LVMs on video reasoning benchmarks. Like Video-LLaVA, ChatGPT-Assistant (version 'gpt-3.5-turbo') is employed to evaluate performance. We directly add `PruMerge` and `PruMerge`+ to Video-LLaVA during inference (without training our own model).

| Methods | LLM size | MSVD-QA | | MSRVT-QA | | ActivityNet-QA | |
|---|---|---|---|---|---|---|---|
| | | Accuracy | Score | Accuracy | Score | Accuracy | Score |
| FrozenBiLM | 1B | 32.2 | - | 16.8 | - | 24.7 | - |
| VideoChat | 7B | 56.3 | 2.8 | 45.0 | 2.5 | - | 2.2 |
| LLaMA-Adapter | 7B | 54.9 | 3.1 | 43.8 | 2.7 | 34.2 | 2.7 |
| Video-LLaMA | 7B | 51.6 | 2.5 | 29.6 | 1.8 | 12.4 | 1.1 |
| Video-ChatGPT | 7B | 64.9 | 3.3 | 49.3 | 2.8 | 35.2 | 2.7 |
| Video-LLaVA | 7B | 70.7 | 3.9 | 59.2 | 3.5 | 45.3 | 3.3 |
| Video-LLaVA + `PruMerge` | 7B | 71.1 | 3.9 | 58.4 | 3.5 | 48.3 | 3.4 |
| Video-LLaVA + `PruMerge`+ | 7B | 71.1 | 3.9 | 59.3 | 3.6 | 47.7 | 3.4 |

Table 4: Token Sampling Strategy Analysis on Different Tasks.

| Approach | #Visual Tokens | Performance |
|---|---|---|
| Task: VQA$^T$ | | |
| LLaVA-PruMerge | 40 | **54.00** |
| Sequential | 40 | 42.72 |
| Spatial | $5 \times 8 = 40$ | 46.85 |
| | $8 \times 5 = 40$ | 47.42 |
| Task: MME | | |
| LLaVA-PruMerge | 40 | **1250.07** |
| Sequential | 40 | 703.60 |
| Spatial | $5 \times 8 = 40$ | 1180.23 |
| | $8 \times 5 = 40$ | 1142.32 |
| Task: POPE | | |
| LLaVA-PruMerge | 35 | **76.2** |
| Sequential | 35 | 11.7 |
| Spatial | $5 \times 7 = 35$ | 69.8 |
| | $7 \times 5 = 35$ | 71.1 |
| | $6 \times 6 = 36$ | 67.9 |
| Task: SQA$^I$ | | |
| LLaVA-PruMerge | 16 | **68.07** |
| Sequential | 16 | 64.20 |
| Spatial | $4 \times 4 = 16$ | 66.29 |

Table 5: Ablation Studies for Adaptive Important Token Selection (AITS, Sec. 3.2) and Token Supplement (TS, Sec. 3.3). With these modules, the downstream performance can be progressively improved.

| Method | LLM | SQA$^I$ | VQA$^T$ | POPE | MME |
|---|---|---|---|---|---|
| LLaVA-1.5 | Vicuna-7B | 66.8 | 58.2 | 85.9 | 1510.7 |
| LLaVA-1.5 w. AITS | Vicuna-7B | 66.5 | 54.8 | 75.7 | 1221.6 |
| LLaVA-1.5 w. AITS & TS | Vicuna-7B | **68.5** | **56.0** | 76.3 | **1350.3** |

Table 6: Ablation on training free and fine-tuning for our approach. With fine-tuning, the performance of LLaVA-PruMerge can be further enhanced.

| Method | LLM | SQA$^I$ | VQA$^T$ | POPE | MME |
|---|---|---|---|---|---|
| LLaVA-1.5 | Vicuna-7B | 66.8 | 58.2 | 85.9 | 1510.7 |
| LLaVA-PruMerge w.o. LoRA-FT | Vicuna-7B | 68.0 | 54.0 | 76.2 | 1250.1 |
| LLaVA-PruMerge w. LoRA-FT | Vicuna-7B | **68.5** | **56.0** | **76.3** | **1350.3** |

**LLaVA-PruMerge**: Our approach dynamically samples key visual tokens (see Sec. 3.2), which results in 40 visual tokens per image on average for TextVQA/MME, 35 tokens for POPE, and 16 tokens for SQA. The visualization is shown in Figure 4 (b).

**Sequential sampling**: We sample $N$ tokens in the flatted visual tokens; e.g., the first 40 tokens are sampled for an apples-to-apples comparison, Fig. 4 (c).

**Spatial sampling**: The sampled $N$ tokens are evenly distributed across the image, Fig. 4 (d-h). We study diverse settings, including $6 \times 6$ (36 tokens), $5 \times 8$ (40 tokens), $8 \times 5$ (40 tokens), $5 \times 7$ (35 tokens), $7 \times 5$ (35 tokens), and $4 \times 4$ (16 tokens).

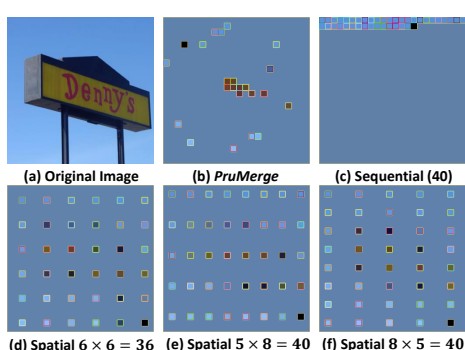

(a) Original Image  (b) *PruMerge*  (c) Sequential (40)

(d) Spatial $6 \times 6 = 36$  (e) Spatial $5 \times 8 = 40$  (f) Spatial $8 \times 5 = 40$

Figure 4: Different token sampling strategies.

Note that all the experiments are done via a training-free manner. As shown in Table 4, our approach is consistently better than sequential sampling and spatial sampling across all downstream tasks, which demonstrates the effectiveness of the sampling mechanism of LLaVA-PruMerge. Importantly, we observe that LLaVA-PruMerge shows much better performance on TextVQA [Singh et al., 2019]. Such Optical Character Recognition (OCR) task requires detailed information about the text, which demonstrates that LLaVA-PruMerge extracts the key information in the images with enough details. This quantitative result aligns with the visualization of LLaVA-PruMerge attentive tokens in Figure 1b, where more attentive tokens are distributed on the foreground text in the images.

### 4.4.2 EFFECTIVENESS OF EACH MODULE IN PRUMERGE

Here, we study the effectiveness of each module in our design based on LLaVA-1.5. Note that we maintain the same amount of visual tokens (6.9%, 40 tokens) across all settings. As shown

Table 7: Comparison with SoTA token reduction methods. Ratio denotes the proportion of remaining tokens. CrossGet results are directly from its paper.

| Method | Ratio | VQA$^{v2}$ | SQA$^I$ | VQA$^T$ | POPE | MME | MMB |
|---|---|---|---|---|---|---|---|
| LLaVA-1.5 | 100% | 78.5 | 66.8 | 58.2 | 85.9 | 1510.7 | 64.3 |
| LLaVA-1.5 + ToMe | 25% | 66.0 | 62.7 | 56.0 | 51.0 | 1385.2 | 56.9 |
| LLaVA-1.5 + ATS | 25% | 66.7 | 63.0 | 55.1 | 57.4 | 1313.2 | 54.9 |
| LLaVA-1.5 + EViT | 25% | 65.5 | 64.1 | 54.2 | 60.1 | 1299.3 | 56.2 |
| LLaVA-1.5 + PruMerge+ | 25% | 76.8 | 68.3 | 57.1 | 84.0 | 1462.4 | 64.9 |
| LLaVA-1.5 + CrossGet$^\star$ | 50% | 77.3 | 66.7 | 54.9 | 83.9 | 1510.2 | 64.7 |
| LLaVA-1.5 + PruMerge+$^\star$ | 50% | 77.6 | 68.5 | 57.6 | 85.1 | 1507.1 | 64.9 |

in Table 5, after progressively adding the proposed modules, including Adaptive Important Token Selection (AITS) and Token Supplement (TS), the downstream performance can be further enhanced.

### 4.4.3 TRAINING ANALYSIS: TRAINING-FREE V.S. FINE-TUNING

Finally, LLaVA-PruMerge can be conducted in either a training-free or fine-tuning manner. With fine-tuning, the large language model can adapt to the new structure of visual tokens, which could further enhance the performance on vision-language tasks. As shown in Table 6, with fine-tuning, our approach does bring better performance for diverse tasks, including ScienceQA [Lu et al., 2022], TextVQA [Singh et al., 2019], POPE [Li et al., 2023b], and MME [Fu et al., 2023].

### 4.4.4 COMPARISON TO TOKEN REDUCTION METHODS

To evaluate the effectiveness of our approach, we compare PruMerge+ with SoTA token reduction methods in the context of Large Multimodal Models. We utilize the token reduction benchmarking framework from Haurum et al. [2023] to implement and compare these methods. Based on the uni-modal token reduction benchmark, ToMe [Bolya et al., 2023], ATS [Fayyaz et al., 2022], and EViT [Liang et al., 2022] are recognized as top performers in uni-modal token reduction. We also include CrossGet [Shi et al., 2024], a concurrent multimodal token reduction method, for comparison. Table 7 clearly demonstrates that PruMerge+ significantly outperforms these unimodal token reduction methods on multimodal tasks, supporting our assertion about its superior effectiveness in managing the complexities of multimodal contexts. Notably, PruMerge+ outperforms CrossGet [Shi et al., 2024], a concurrent multimodal token reduction method, under the same reduction ratio.

**Advantages over uni-modal token merging methods:** The superior performance of PruMerge+ on multimodal tasks can be attributed to several key factors: **(1) Sparsity-based selection:** PruMerge leverages the sparsity observed in multimodal encoders (Visual-LLM and Video-LLM), particularly in how attention scores distribute sparsely in the final layer. This pattern is less pronounced in unimodal token reduced models where token relevance may not distribute in the same way. **(2) Efficiency in LMMs:** Considering that the bulk of computational demand within LMMs is attributed to the LLM backbone rather than the modality encoder, our approach focuses not only on the reduction of tokens but also on maximizing the informational content of the pruned visual tokens. In contrast, unimodal token merging methods focus on the modality encoder's internal efficiency. This strategy addresses the computational challenges inherent in LMMs with minimal compromise in the quality of the visual representation. **(3) Accumulation phenomenon:** PruMerge capitalizes on the accumulation of sparsity across layers, a characteristic more specific to multimodal models. Unimodal token reduction methods that reduce tokens gradually cannot leverage this sparsity effectively.

## 5 CONCLUSION

In this paper, we improve the efficiency of Large Multimodal Models (LMMs) from the perspective of reducing the quantity of visual tokens. By leveraging the spatial redundancy in visual tokens, we proposed a plug-and-play token reduction module that employs the similarity between the class token and spatial tokens as a key criterion for pruning and merging visual tokens. Our approach, applied to LLaVA-1.5, demonstrated that by utilizing only 6.9% of visual tokens on average, the pruned tokens can maintain comparable performance across a wide range of visual question-answering and reasoning tasks. Notably, our work highlights the potential for significant computational savings without sacrificing the reasoning capabilities of LMMs. We hope our work inspires further exploration into the interplay between efficiency and performance in LMMs.

# 6 REPRODUCIBILITY STATEMENT

We have provided the code of `PruMerge` and `PruMerge+` algorithms as supplementary material. The code has been anonymized. Additionally, we intend to publicly release the code, data, pretrained models, and any other resources necessary for the community to fully reproduce our work.

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

# A  APPENDIX

## A.1  SOCIETAL IMPACTS

In this study, we propose a technique that improves the efficiency of Large Multimodal Models (LMMs), making them more accessible. This approach helps to democratize LMMs by lowering deployment costs and hardware barriers, facilitating their use in edge computing. However, it does not mitigate the potential misuse of LMMs by malicious actors.

