# OpenReview forum: "LLaVA-PruMerge: Adaptive Token Reduction for Efficient Large Multimodal Models"
_ICLR.cc/2025/Conference — ICLR 2025 Conference Withdrawn Submission_

### Official Review · Reviewer_6QgK · 2024-10-28

**Soundness:** 2
**Presentation:** 3
**Contribution:** 2
**Rating:** 3
**Confidence:** 3

**Summary:**

The paper presents PruMerge, a method designed to improve the computational efficiency of large multimodal models (LMMs) by adaptively reducing visual tokens. Utilizing sparse attention distribution from the visual encoder, PruMerge selects important tokens, achieving comparable performance to full models with fewer tokens on LLaVA.

**Strengths:**

1.Enhanced Computational Efficiency: PruMerge reduces visual tokens, decreasing computational complexity and speeding up large multimodal models, making them more suitable for resource-constrained environments.

2.Adaptive Selection Mechanism: By leveraging sparse attention, the method retains essential visual information while reducing computation, allowing for flexibility based on the complexity of input images.

**Weaknesses:**

1.A key limitation of this method is that, by trimming visual tokens, it risks losing essential image details. This makes it less suited for tasks that rely heavily on fine-grained visual information, like OCR, object detection, or fine-grained classification. Since these tasks require precise visual cues to perform well, reducing tokens could hurt performance. Another issue is that the paper doesn't test on these detail-sensitive tasks, leaving some doubt about the method's broader applicability.

2.A key limitation is that experiments were only conducted on LLaVA-based models, limiting insights into the method’s generalizability. Testing solely on one model type makes it unclear if the token-pruning benefits would hold across different multimodal architectures. Broader testing would provide a fuller picture of its effectiveness.

3.The experimental section lacks clarity regarding whether the reported results are based on the pre-fine-tuned model or the fine-tuned version. This ambiguity can lead to confusion about the actual effectiveness of the proposed method and its reliance on fine-tuning.

**Questions:**

1.Could the authors clarify if they have tested this method on tasks that require high levels of visual detail, such as image captioning? If so, how did the method perform in comparison to the non-pruned baseline?

2.Have the authors explored whether the assumption of visual token redundancy holds across different image complexities, such as high-texture scenes or densely populated images? How does token pruning adapt to varying levels of image detail?

Suggestion:It would be beneficial for the authors to explicitly state whether the results are obtained from the fine-tuned model or the base model. Additionally, providing a comparison between both scenarios would enhance the transparency of the experiments and allow readers to better assess the method's true performance and its dependence on fine-tuning.

---

### Official Review · Reviewer_FRdG · 2024-10-28

**Soundness:** 3
**Presentation:** 2
**Contribution:** 2
**Rating:** 3
**Confidence:** 4

**Summary:**

This paper introduces LLaVA-PruMerge, a method designed to optimize Large Multimodal Models (LMMs) by reducing the number of visual tokens without compromising performance. The central innovation, the "PruMerge" algorithm, adaptively prunes and merges visual tokens based on their relevance, addressing the computational inefficiencies typically associated with LMMs. By exploiting sparsity patterns in attention scores between class and visual tokens, PruMerge reduces token counts by up to 94% on average. Empirical results demonstrate that this method improves efficiency while maintaining performance across diverse visual question-answering and reasoning benchmarks, including VQAv2 and ScienceQA. The approach has been evaluated on both image and video models, highlighting its adaptability in optimizing LMMs across different modalities.

**Strengths:**

- High Visual Token Pruning Rate: The proposed method achieves a significant reduction in computational cost for Multimodal Large Language Models (MLLMs) by pruning a high proportion of visual tokens. This makes the model more efficient in resource-constrained settings.
- Inference Speed and Memory Efficiency: By pruning visual tokens before they are fed into the LLM, the approach reduces both inference time and memory usage. This aspect is particularly advantageous for practical applications where memory efficiency is crucial.

**Weaknesses:**

- Potential Loss of Auxiliary Visual Information: The aggressive pruning of visual tokens could lead to a significant loss of auxiliary information, which is not evident from experiments on the VQA dataset alone. This likely contributes to weaker performance on the POPE dataset. Given the request-agnostic nature of pruning method , it may only preserve the primary content, limiting its broader applicability. Testing the method on tasks requiring richer image information, like image captioning, would better showcase its robustness.

- Performance Degradation Despite High Pruning Rate: Although the method achieves a high pruning rate, it incurs substantial performance drops, approximately 8% with prumerge and 2% with prumerge+, which is considerable, especially when compared to other token pruning approaches like FastV [1] and VTW [2] , which maintain performance integrity. Given the extra fine-tuning required, this performance decrease could be prohibitive in practical applications.

- Limited Memory Savings in Long Text Generation: While the proposed method reduces memory usage during inference, the focus on visual token pruning alone may have minimal impact for long text generation tasks, where large numbers of language tokens drive memory consumption. The memory savings from visual token reduction are less impactful in these contexts. Furthermore, in short text generation tasks, where accuracy and speed are primary concerns, the proposed method does not excel.

[1] Chen, Liang, et al. "An image is worth 1/2 tokens after layer 2: Plug-and-play inference acceleration for large vision-language models." arXiv preprint arXiv:2403.06764 (2024).
[2] Lin, Zhihang, et al. "Boosting Multimodal Large Language Models with Visual Tokens Withdrawal for Rapid Inference." arXiv preprint arXiv:2405.05803 (2024).

**Questions:**

- Consistency of Results in Table 1 and Table 6: Are the results reported in Table 1 based on additional fine-tuning? Table 6 shows that fine-tuning has a positive effect on performance, so confirming whether Table 1 results include fine-tuning would clarify the impact of this step on the final results, ensuring data consistency.

- Comparison with Visual Token Pruning Methods like FastV [1] and VTW [2] : Although FastV and VTW conduct token pruning within specific layers of the model, their goals align closely with the proposed method in terms of reducing computational cost through pruning. Since these methods reportedly maintain model performance with minimal loss, a comparison with them would provide a clearer reference, especially in terms of balancing accuracy with computational efficiency.

- Applicability Beyond VQA to More General Scenarios: VQA tasks primarily require only the main content of an image, allowing aggressive pruning strategies without significant impact on performance. However, in multi-turn dialogues with large language models, preserving comprehensive image information is essential, as responses may require details beyond the main content. Could you demonstrate the method’s performance on image captioning tasks, where capturing detailed information is crucial? This would better showcase its efficacy in tasks that require a fuller representation of visual data.

[1] Chen, Liang, et al. "An image is worth 1/2 tokens after layer 2: Plug-and-play inference acceleration for large vision-language models." arXiv preprint arXiv:2403.06764 (2024).
[2] Lin, Zhihang, et al. "Boosting Multimodal Large Language Models with Visual Tokens Withdrawal for Rapid Inference." arXiv preprint arXiv:2405.05803 (2024).

---

### Official Review · Reviewer_9ejG · 2024-10-30

**Soundness:** 3
**Presentation:** 3
**Contribution:** 2
**Rating:** 5
**Confidence:** 4

**Summary:**

In this paper, authors propose a training-free token reduction strategy, PruMerge, for efficient MLLM inference. The method focuses on sampling the most important visual tokens by outlier token detection and clustering. PruMerge manages to reduce the computational costs by approximately 10 times in FLOPs with comparable performance.

**Strengths:**

1) The method effectively chooses the most informative tokens from the visual encoder without further finetuning, hence largely reducing the computation costs of MLLM.

2) PruMerge outperforms former SOTA methods on token pruning for ViTs.

**Weaknesses:**

Instead of being an adaptive reduction method for MLLM, PruMerge seems more like a strategy for the transformer-based visual encoder. It makes no exploration of the design of the LLM, particularly, ignoring the interaction between visual tokens and text tokens in MMLM.  If so, authors may consider exploring PriMerge on more visual encoders, e.g., SigLIP widely used in MLLMs and etc.

Also, as authors claim that token relevance matters more for MLLM encoder, the paper may support it with more experimental exploration as well as visualizations.

**Questions:**

As PruMerge is effective for LLaVA1.5, can it be adapted for MLLMs with AnyRes (which is a widely adopted strategy), e.g., LLaVA-Next?

---

### Official Review · Reviewer_tbBz · 2024-11-02

**Soundness:** 2
**Presentation:** 3
**Contribution:** 2
**Rating:** 3
**Confidence:** 5

**Summary:**

This paper introduces PruMerge, a method to improve the efficiency of Large Multimodal Models by reducing their visual token count. Specifically, PruMerge address the computing efficiency with two steps: 1) it first uses an Interquartile Range method to identify visual tokens outliers based on attention scores, 2) and then applies k-nearest neighbor clustering to merge similar tokens. The authors also introduce PruMerge+, which adds spatially uniform sampling. Experiments on LLaVA-1.5 show that PruMerge achieves comparable performance across various visual reasoning tasks while using only 5.5% of the original tokens. The method is also showed to be effective with video LMMs, reducing Video-LLaVA's 2048 tokens to 256 tokens (12.5%) without compromising the performance.

**Strengths:**

1. The proposed method provides a new solution towards multimodal LLM efficiency in the direction of input compression. The proposed method is able to provide computation reduction without significant finetuning.

2. Experiments show the proposed method is effective, which can maintains performance using only 5.5% of original tokens in LLaVA-1.5 across six benchmarks. It can also reduces Video-LLaVA's tokens from 2048 to 256 while preserving or improving performance.

3. The proposed method doesn't requiring model architecture change, makes it easier to deploy.

4.. The paper writing is easy to understand.

**Weaknesses:**

1. The experiments are too limited to show the generalization ability of the proposed approach. The method is primarily validated on LLaVA-1.5 and Video-LLaVA. Additional experiments on other LMMs like Flamingo, Qwen, etc. would better show the method's generalizability across different MLLM modeling architectures like different image encoder and different feature fusion mechanism.

2. The experiment section missed real world efficiency analysis. The results on Tesla V100 GPU (e.g. in Table 2) doesn't represent typical deployment for efficient LMMs on resource-constrained environments (e.g. on mobile device).  For example, the paper would be more solid if the authors could show the memory, FLOPs, and latency on mobile device like Macbook, or Phone (e.g. iPhone or Pixel).

3. The experiment missed important baselines. While the paper compares with ToMe, ATS, and EViT, it misses several crucial token reduction baselines such as Token Pruning[1], Token summarization[2], Progressive Token Pruning[3], and recent prompt compression methods. These baselines are the latest work that achieved better performance over the aforementioned ones in various tasks.

4. The paper is based on the assumption of sparsity in visual encoder (CLIP). However, the paper doesn't check whether this holds true for other visual encoders (i.e. different encoder architectures like CNN and ViT).

5. The requirement of attention scores for outlier selection makes this method incompatible with FlashAttention, potentially leading to higher memory usage and increased latency. The authors should provide comparisons with FlashAttention-enabled baselines to address deployment concerns. Specifically, peak memory consumption and latency measurements comparing PruMerge/PruMerge+ vs FlashAttention-enhanced baselines are needed to verify the effectiveness of the proposed method.


[1] https://arxiv.org/abs/2403.06764
[2] https://arxiv.org/abs/2410.14072
[3] https://arxiv.org/abs/2301.13741

**Questions:**

Refer to the "Weakness" part.

---

### Note · Authors · 2024-11-12

I have read and agree with the venue's withdrawal policy on behalf of myself and my co-authors.